# Investigation of Cell Damage Induced by Silver Nanoparticles in a Model Cell System

**DOI:** 10.3390/pharmaceutics17040398

**Published:** 2025-03-21

**Authors:** Sergey Pirutin, Dmitrii Chaikovskii, Mikhail Shank, Mikhail Chivarzin, Shunchao Jia, Alexander Yusipovich, Oleg Suvorov, Yuehong Zhao, Dmitry Bezryadnov, Andrey Rubin

**Affiliations:** 1Faculty of Biology, Shenzhen MSU-BIT University, Shenzhen 518172, China; pirutin@smbu.edu.cn (S.P.); 1220220033@smbu.edu.cn (M.S.); 6620220043@smbu.edu.cn (M.C.); jsc@smbu.edu.cn (S.J.); rubin@biophys.msu.ru (A.R.); 2Faculty of Biology, Lomonosov Moscow State University, Moscow 119991, Russia; yusipovich@biophys.msu.ru (A.Y.); tiunovaaa@my.msu.ru (D.B.); 3Institute of Theoretical and Experimental Biophysics of Russian Academy of Sciences, Pushchino 142290, Russia; 4Institute of Food Systems and Health-Saving Technologies, Russian Biotechnological University, Moscow 125080, Russia; suvorovoa@mgupp.ru; 5School of Medicine, The Chinese University of Hong Kong, Shenzhen 518172, China; zhaoyuehong@cuhk.edu.cn

**Keywords:** silver nanoparticles, membrane damage, peritoneal macrophages, reactive oxygen species, lipid peroxidation, mathematical modeling, oxidative stress, nanotoxicology

## Abstract

**Background/Objectives**: The growing diversity of novel nanoparticle synthesis methods, particularly for silver nanoparticles (AgNP), coupled with their significant biological activity and wide range of applications across various medical fields, necessitates a comprehensive investigation into the consequences of particle-induced cellular damage. This study aimed to investigate AgNP-induced damage to macrophage plasma membranes, focusing on concentration, temperature, incubation time, and the role of pro- and antioxidant factors, using model systems based on mouse peritoneal macrophages. **Methods:** Mouse peritoneal macrophages were incubated with AgNP (0.1–10 μg/mL) at temperatures ranging from 4 °C to 37 °C. Membrane integrity was assessed via microfluorimetric analysis. The influence of prooxidant (UV-B) and antioxidant (serotonin) factors was also examined. A mathematical model was developed to describe the interaction between AgNP and macrophages. **Results:** The diameter of our synthesized silver nanoparticles, assessed via dynamic light scattering (DLS), ranged from 5 to 170 nm, with a predominant size distribution peak at 70 nm. AgNP caused dose- and temperature-dependent membrane damage, which was more pronounced at 4 °C and 37 °C than at 22 °C and increased with incubation time. UV-B enhanced membrane damage, while serotonin mitigated it. The mathematical model correlated strongly with the experimental data, emphasizing the role of ROS in membrane disruption. AgNP also dose-dependently increased ROS generation by macrophages. **Conclusions:** AgNP, in doses of 0.1–10 μg/mL, induces dose-dependent membrane damage in macrophages. The developed model is a useful tool for predicting nanoparticle toxicity. Together with the experimental findings, it highlights the critical role of ROS, lipid peroxidation, the lipid bilayer state, and antioxidant defenses in AgNP-induced membrane damage.

## 1. Introduction

The modern nanotechnology industry is rapidly developing and introducing new products into our lives with unique properties and sizes ranging from 1 to 100 nm [1,2,3,4,5,6,7]. Due to the small size of nanoparticles, a large proportion of their structural units are on the surface, which determines the special physical and chemical properties of such particles [6,8,9,10,11]. These properties grant most nanoparticles significant biological activity and enable their widespread application in various fields of medicine [11,12].

Among the vast number of synthesized nanomaterials, silver nanoparticles (AgNP) are of particular interest due to their extensive antimicrobial activity and the absence of resistance in most pathogenic microorganisms [13,14,15,16,17,18,19,20]. They are also considered to have relatively low toxicity toward eukaryotic cells [21]. However, according to the literature, the biological properties of AgNP are ambiguous and depend on their size, their synthesis methods, and the conditions of their action on biological targets [11,22,23,24]. In some cases, AgNP exhibit toxicity to living organisms at various levels of organization [12,18,23,25,26,27,28,29]. Therefore, it is justifiable to increase the number of studies on the toxicity of AgNP at the cellular level.

In addition to the size of AgNP, as mentioned above, their biological activity is influenced by their shape, duration of exposure, concentration, and method of synthesis [15,22,30,31,32]. Various researchers have noted that a significant portion of the toxic effects of AgNP is attributed to the generation of reactive oxygen species (ROS) [18,31]. Some authors suggest that silver ions (Ag^+^) act as an intermediate between AgNP and ROS formation [7,31,33,34,35,36]. Other authors propose that ROS are directly generated by AgNP [29,37,38,39,40]. In any case, the ROS generated by nanoparticles can damage biological membranes and DNA, causing necrosis or apoptosis in cells [11,31,33,41,42,43,44,45,46,47,48,49]. In cell membranes, the lipid bilayer is particularly vulnerable to ROS, which intensify the process of lipid peroxidation (LPO), leading to defects in the lipid matrix and consequently to membrane integrity disruption [11,50,51]. The resistance of the lipid bilayer to damage due to intensified LPO depends on factors such as temperature and incubation conditions, the presence of exogenous pro- and antioxidants, and the state of the cell’s antioxidant defense (AOD) system [52,53,54,55,56,57].

Thus, the manifestation of the toxicity of nanoparticles is a complex process dependent on many factors and requires an in-depth analysis and understanding of the mechanisms of nanoparticles’ action on biological targets [11]. Research accounting for this multitude of factors may require a comprehensive and labor-intensive approach with significant financial costs [23]. Therefore, it is important to supplement experiments on model systems with the creation of various theoretical models that will serve as additional important tools for predicting nanoparticle toxicity and testing hypotheses about the mechanisms of such toxicity [58].

Experiments on model cell systems are essential in these studies to understand the mechanisms of nanoparticle-induced cell damage [30,59,60,61,62,63,64,65,66,67,68,69,70,71]. We believe that cell preparations based on phagocytic cells can serve as a convenient model system for such experiments, as these cells play a crucial role in the immune system and are capable of providing a rapid functional response to various physico-chemical factors, including nanoparticles [72,73,74].

Given the above, the aim of our study was to investigate the effect of AgNP on the plasma membranes of cells at micromolar concentrations at varying temperatures and incubation times and in the presence of pro- and antioxidant factors using cell model systems based on peritoneal mouse macrophages. Additionally, we aimed to develop a theoretical mathematical model that takes into account the interaction of AgNP and cells under different temperature conditions. As part of the study objectives, we investigated the effects of silver nanoparticles on macrophage-generated ROS and their modulation of LPO intensity in macrophage membranes

## 2. Materials and Methods

### 2.1. Object of Study

The object of the study was mouse peritoneal macrophages. Cell preparations were obtained using the standard peritoneal lavage technique [75] with a buffer solution based on Hank’s solution supplemented with 10 mmol/L HEPES (pH 7.2) (SERVA Electrophoresis, Heidelberg, Germany).

The isolated suspension contained 95% macrophages. The concentration of the cell suspension was adjusted to 1.0 × 10^6^ cells/mL using the same solution. Then, 30 microliters of the resulting cell suspension were applied to coverslips and incubated at 22 °C in a humid chamber for 45 min to allow for macrophages to adhere to the coverslips. After incubation, the coverslips were washed with Hank’s solution with HEPES to remove non-adherent cells and placed in plastic Petri dishes containing 2 mL of the same solution. The cells were maintained under these conditions after isolation and throughout the experiment. This method of isolating macrophages and maintaining them in the given incubation medium during the experiment is optimal for preserving macrophages in an unstimulated native form, avoiding pre-stimulation, known as priming, which is a crucial aspect of our model approach. It is noteworthy that the cell isolation protocols and sample preparation methodology maintained macrophages in a rounded morphological state throughout the experimental timeline. This conserved spherical morphology served as a foundational geometric parameter in the development of our mathematical model. Furthermore, the maximum experimental duration of 3.5 h did not compromise macrophage viability or functional integrity in control samples incubated under identical conditions. The pH of the incubation medium was continuously monitored and maintained at a constant level of 7.2 throughout the study. Membrane integrity in control groups was systematically assayed during the entire experimental timeframe. Control samples were defined as untreated cell preparations incubated at ambient laboratory temperature (22 °C).

### 2.2. Method for Determining Plasma Membrane Integrity

To study the toxicity of silver nanoparticles, they were added to the cell incubation medium at final concentrations ranging from 0.1 to 10 µg/mL. The incubation time of cells with nanoparticles varied from 5 to 210 min. The incubation medium temperature varied from 4 °C to 37 °C, with specific temperatures of 4, 15, 22, and 37 °C.

The content of cells with damaged plasma membranes was analyzed using the Axio Imager Z2 ZEISS fluorescent direct microscope (Carl Zeiss Microscopy, Jena, Germany). The integrity of the macrophage plasma membranes was determined via dual staining with two fluorescent dyes, ethidium bromide (EB) and fluorescein diacetate (FDA), at a final concentration of 5 µg/mL (SERVA Electrophoresis, Heidelberg, Germany). The cells were incubated with fluorochromes for 5 min; then, the damaged cells were counted after removing the unbound dye by washing the cell preparations once with the original incubation medium. Cells with intact plasma membranes exhibited green fluorescence of polar fluorescein molecules, which accumulated inside the cytoplasm by cleaving acetate tails from nonpolar FDA molecules through cellular esterases. These nonpolar FDA molecules easily penetrate the cytoplasm through the lipid bilayer of the cell membrane due to their electro-neutrality [76]. When the cell membrane is compromised, the accumulated polar fluorescein molecules exit the cytoplasm into the incubation medium, while EB molecules enter the cell and bind to nucleic acids in the nucleus [77]. Thus, cells with damaged membranes are identified by the red-orange fluorescence of their nuclei. Green and red fluorescence were observed in two channels using fluorescent microscope filter cubes: ZEISS Filter Set 38 (BR470/40, FT495, BR525/50) and ZEISS Filter Set 20 (BR546/12, FT560, BR575-640). Fluorescent emissions from the samples were induced using irradiation from a mercury lamp—X-cite 012-63000 illuminator—X-cite 120Q, which is part of the used microscope.

The toxicity of nanoparticles was assessed using the relative content of cells with damaged membranes in randomly selected microscopic fields of view. Counting was performed in at least 50 fields, with a minimum of 1500 cells in each cell preparation. The integrity of macrophage membranes in the control was monitored throughout the entire duration of the experiment. Control samples were defined as untreated cell preparations incubated at ambient laboratory temperature (22 °C). Experiments were conducted in at least three independent series, with three repetitions in each series. The obtained data are presented as the mean values and standard deviation (mean ± standard deviation) of the studied parameters at different times and concentrations.

### 2.3. Cell Incubation Conditions and Their Modification

For cell incubation at 30 °C and 37 °C, a dry-air thermostat (Wiggens/WH-25, Berlin, Germany) was used. Cell incubation at temperatures of 4 °C and 15 °C was performed in refrigerated chambers. The standard incubation medium temperature was considered to be room temperature (22 °C), maintained by the room’s climate control system.

The incubation medium was modified by adding serotonin to a final concentration of 1 mg/mL two minutes before the addition of AgNP.

The temperature range of 4–37 °C was selected for two principal reasons. First, enzymatic processes—particularly the activity of antioxidant enzymes—are markedly suppressed at 4 °C, while 37 °C represents a physiologically relevant temperature. Second, this interval encompasses two phase transitions (gel to liquid crystalline states) in the lipid bilayers of most biological membranes. These structural reorganizations are accompanied by altered bilayer resistance to LPO processes.

### 2.4. Method of Synthesizing AgNP

A colloidal solution of AgNP was prepared using a modified method described in [78]. A total of 45 mL of 1.5 × 10^−3^ M hydroxylamine hydrochloride solution was added to a 200 mL round-bottom flask at room temperature (22 °C). This solution was then adjusted to pH 10 using a 1 M NaOH solution. While stirring constantly with a magnetic stirrer at a speed of 200 rpm, 100 µL portions of a 10^−2^ M AgNO_3_ solution were added to a total volume of 1 mL. After the addition of AgNO_3_ to the flask, the solution was stirred for an additional 10 min until it reached an orange-yellow color with a greenish tint. The resulting solution had a concentration of 2 × 10^−4^ M (20 mg/mL) and was stored at ambient temperature. In our experimental framework, we utilized concentration units of mg/mL and µg/mL, as this is the most prevalent metric in comparable biological studies [79,80]. The accelerated stability assessments demonstrated that the nanoparticle size distribution remained invariant for storage periods exceeding one month under ambient conditions (22 °C), with no detectable particle aggregation phenomena observed during this timeframe.

### 2.5. UV-Vis Absorption Spectroscopy Method

Using the method of absorption spectroscopy, the absorption spectra of the synthesized AgNP suspensions were obtained. For this purpose, a dual-beam cuvette spectrophotometer with an integrating sphere, UV-Vis Shimadzu UV-2600 (Shimadzu Corporation, Kyoto, Japan), and standard quartz cuvettes (Hellma & Co. KG, Müllheim, Germany) with a 10 mm optical path length were used. The absorption spectra were measured using AgNP suspension solutions in the concentration range from 0.5 to 20 µg/mL.

### 2.6. Scanning Electron Microscopy (SEM)

Using scanning electron microscopy, microphotographs of fixed macrophages were obtained for the control and after a 60 min incubation with nanoparticles at a final concentration of 5 μg/mL. Fixed macrophage preparations were created according to a standard protocol [81]. The cell-containing samples were fixed in 10% neutral formalin and washed in phosphate-buffered saline (Paneco, Moscow, Russia). Dehydration was then performed by successive treatment with increasing concentrations of ethanol. Finally, hexamethyldisilazane (EKOS-1, Moscow, Russia) was added to the samples, and after 30 min the liquid was completely removed, with the samples subsequently dried under an exhaust hood at room temperature.

After fixation, the samples were placed in the vacuum chamber of the scanning electron microscope KYKY-EM6200 (KYKY Technology Co., Ltd., Beijing, China). Once the vacuum in the chamber exceeded 3 × 10^−5^ Torr, the samples were scanned and photographed. The microscope settings and image acquisition were controlled using the proprietary software KYKY-EM6200 Version 1.5.0.3. The main microscope parameters were set as follows: filament saturation current of 2.5 A and high voltage of 15 kV, with magnifications ranging from 4.70 × 10^3^ to 10.00 × 10^3^.

### 2.7. Transmission Electron Microscopy (TEM)

An analysis of the sizes and shapes of nanoparticles was also performed using the TEM method based on the obtained images. Sample preparation followed a modified method [82]. A drop of AgNP suspension (10 µL) was applied to a copper grid substrate (150 mesh), coated with formvar and a carbon film (approximately 10 nm). After the applied material dried on the substrate, the prepared sample was placed in the chamber of a JEOL JEM-1011 electron microscope (JEOL, Tokyo, Japan). The microscope used a tungsten cathode with an accelerating voltage of 80 kV. Images were captured using a Gatan ORIUS SC1000W (model 832) CCD digital camera with a resolution of 4008 × 2672 and original software—Digital Micrograph (GATAN). The magnification of the microscope was 230,000×.

### 2.8. Dynamic Light Scattering Method (DLS)

The size analysis of AgNP was conducted using the Litesizer 500 Particle Analyzer (Anton Paar Corporation, Graz, Austria), which determines dynamic light scattering based on accounting for fluctuations in the random motion of particles in the sample, thereby determining the average particle diameter [83,84]. The Litesizer measures the average diameter as the hydrodynamic diameter of a sphere that has the same diffusion coefficient as the particle being studied. The intensity of light oscillation depends on the diffusion coefficient, which in turn depends on particle size. Thus, the instrument determines the hydrodynamic diameter of the particles using the Stokes–Einstein equation:$$D_{H}=\frac{kT\;}{3\pi \eta D}$$
where k is the Boltzmann constant, T is the absolute temperature, and η is the viscosity of the solution.

The setup used a 40 mW semiconductor laser with a wavelength of 658 nm and the following characteristics: measurement range ≥ 1000 mV; mobility range from 10^−11^ to 2 × 10^−7^ m^2^/s. The instrument performed 1000 scans at a temperature of 22 °C. The obtained data were processed using the original software Kalliope Version 1.2.0 (Anton Paar Corporation, Graz, Austria).

### 2.9. Mathematical Model of AgNP Interaction with Macrophages

To describe the kinetics of the interaction between silver nanoparticles and cells, a mathematical model was developed based on the theory of diffusion-controlled reactions and considering the radical physicochemical processes that occur when nanoparticles contact the cell membrane [85,86]. The model is based on the concept of the following two parallel reactions:-First Reaction: The diffusion transfer of nanoparticles to the cell surface and the formation of a “particle–membrane” complex, described within the framework of the Smoluchowski theory for diffusion-controlled processes [87]. To determine the rate constant, an analytical solution of the diffusion equation in spherical coordinates was used, taking into account boundary conditions corresponding to an irreversible reaction on the cell surface.-Second Reaction: This is based on the Arrhenius law, which accounts for the temperature dependence of the reaction [88]. This approach considers the generation of ROS upon contact of nanoparticles with the membrane and the subsequent overall work of the cell’s AOD system using a probabilistic approach. The actions of the AOD system protect potentially vulnerable cellular structures; however, cell death occurs if the AOD system is insufficiently effective, particularly in cases with a significant accumulation of toxic peroxidation products over time.

The kinetics of the change in the concentration of dead cells are described via a first-order ordinary differential equation, accounting for the balance of the rates of the two mentioned reactions. The analytical solution of this equation under the presented initial conditions was used for comparison with experimental data and to determine the model parameters using the method of least squares.

### 2.10. UV-B Irradiation of Cells

The enhancement of the LPO process in cell preparations was achieved by irradiation with medium-wave UV radiation from a 1 kW mercury lamp and an interference filter with a transmission maximum at λ_max_ = 306 nm. The power density of the UV radiation was 1.4 mW/cm^2^, and the radiation dose was 5 J/cm^2^. AgNP was added to the cell incubation medium 1 min before the start of irradiation.

### 2.11. Determination of ROS in Cell Suspensions

The effect of AgNP on intracellular ROS levels in cell suspensions was assessed using a modified method [89] with the fluorescent probe 2′,7′-dichlorodihydrofluorescein diacetate (DCFH-DA, Sigma-Aldrich, St. Louis, MO, USA) at a final concentration of 10 μg/mL. Measurements were performed using a universal plate reader, Thermo Scientific Varioskan LUX (Thermo Fisher Scientific, Waltham, MA, USA).

Freshly isolated peritoneal macrophages at a density of 1.25 × 10^5^ cells/mL in a volume of 25 μL were seeded into wells of black-walled, clear-bottom, 96-well plates (Thermo Scientific™, non-treated, Rochester, NY, USA) and allowed to adapt and settle for 15 min at 22 °C. Subsequently, 175 μL of Hanks’ solution with HEPES containing DCFH-DA was added to the wells, and the plates were incubated in the dark at 22 °C. The plates were then centrifuged at 500 g for 7 min using an Eppendorf™ Centrifuge 5810R (Hamburg, Germany) with plate adapters. After centrifugation, the supernatant was carefully removed, along with any unbound dye. To the settled cells, 200 μL of Hanks’ solution with HEPES containing AgNP at final concentrations of 1, 2.5, and 5 μg/mL, or plain buffer (control), was added.

For positive controls, 200 μL of Hanks’ solution with HEPES containing hydrogen peroxide (final concentration 30 mM) or phorbol myristate acetate (PMA, final concentration 20 nM), a known ROS-inducer via NADPH oxidase activation [90], was added. Each sample, with a specific agent and concentration, was prepared in at least four replicates on each 96-well plate.

The plates were then placed in designated incubation chambers set at 22 °C and 37 °C. After a 10 min temperature adaptation, the first fluorescence measurement of the accumulated ROS-reacted DCFH-DA dye was performed at an emission wavelength of 522 nm with an excitation wavelength of 493 nm. Following a 20 s measurement, the plates were returned to their respective incubation chambers. Subsequent measurements were taken at 15 min intervals for a total duration of 210 min. Samples that did not have any macrophage-stimulating agents added were considered controls.

### 2.12. Statistical Data Processing

The obtained results were statistically treated using a GraphPad Prism 10 for Windows 64-bit Version 10.3.0.507 (GraphPad software, Boston, MA, USA). The data are presented as mean ± standard deviation; the significance of differences between groups was estimated using a one-way ANOVA test with a posterior Tukey criterion (*p* < 0.05).

The difference between the experimental results and forecast in terms of the dependence of the number of damaged cells on the incubation time at different temperatures was calculated using the coefficient of determination (see Section 3.3.4).

## 3. Results and Discussion

### 3.1. Study of the Size of Synthesized AgNP

#### 3.1.1. UV-Vis Absorption Spectroscopy

The absorption spectra of the synthesized AgNP suspension in the concentration range of 0.5–20 µg/mL were investigated using the UV-Vis absorption spectroscopy method. As shown in Figure 1, the absorption maximum corresponds to 403 nm. In the region of the absorption maximum, the optical density increases and varies based on concentration in the range of 0.17–3.43 units. Changes in the concentration do not shift the position of the peak. The position of the absorption spectra maxima of the AgNP suspensions synthesized by us corresponds to the typical position for the absorption spectra maxima of AgNP obtained by other researchers and indicates AgNP sizes in the range of 15–100 nm [91,92,93,94,95]. Using our synthesis method, storing AgNP for over a month at room temperature did not lead to changes in their absorption spectra, indicating a stable state of the AgNP suspension and the absence of aggregation.

Next, using methods such as scanning electron microscopy, transmission electron microscopy, and dynamic light scattering, we obtained detailed characteristics of the parameters of the synthesized nanoparticles.

#### 3.1.2. Transmission Electron Microscopy

More accurate sizes and shapes of the synthesized AgNP are shown in the micrographs obtained using the TEM method. These images were taken with particles deposited on a copper grid substrate coated with formvar and carbon film at a magnification of 230,000×.

As evidenced by Figure 2, the nanoparticles exhibit a polygonal morphology with rounded edges, ranging in size from 10 to 100 nm. For subsequent analyses, we employed a geometric simplification by approximating the nanoparticles as spherical entities.

#### 3.1.3. Determination of AgNP Size by Dynamic Light Scattering

Using dynamic light scattering, a method based on analyzing fluctuations in the random Brownian motion of particles within a sample, we determined the mean hydrodynamic diameter of synthesized AgNP dispersed in cell incubation medium (HEPES-buffered Hanks’ balanced salt solution) at a final concentration of 5 mg/mL. The hydrodynamic diameter was defined as the diameter of a sphere exhibiting an equivalent diffusion coefficient to the investigated AgNP.

As seen in Figure 3, the size distribution of nanoparticle diameters in the suspension has two peaks: the first around 80 nm and the second, constituting less than one percent, around 8 nm. The statistical program in the software associated with the Kalliope instrument determined the average diameter of AgNP in the suspension to be around 70 nm. The mean hydrodynamic diameter of 70 nm, as determined through our characterization studies, was implemented as a critical parameter in our computationally developed mathematical model. Accelerated stability testing confirmed that the nanoparticle size distribution remains invariant for storage periods exceeding one month under ambient conditions, with no detectable aggregation phenomena observed throughout the monitored duration.

### 3.2. Dependence of the Membranotropic Action of AgNP on Concentration, Temperature, and Incubation Time

This part of the work investigates the dependence of AgNP toxicity on its concentration in the cell incubation medium at different environmental temperatures and for different incubation times. Figure 4 shows the dependencies of the damaging effect of AgNP on macrophage cell membranes with particle concentrations in the range of 0.1–10 µg/mL at temperatures of 22 °C and 37 °C.

As seen in Figure 4, at a temperature of 22 °C, a significant and reliable relative number of cells with damaged plasma membranes in the macrophage population begins to appear in the presence of AgNP at a concentration of 2.5 µg/mL. Further increasing the final concentration of AgNP to 10 µg/mL leads to a concentration-dependent increase in the number of cells with damaged membranes in the macrophage population, which reaches approximately 30% at a concentration of 10 µg/mL. Increasing the incubation temperature to 37 °C with the same AgNP concentrations results in a more pronounced toxic effect, with around 20% of damaged cells observed even at the lowest tested concentration (0.1 µg/mL). The dependence of the damaging effect on macrophage plasma membranes on concentration follows a pronounced S-shaped curve. The highest concentration of AgNP in our study (10 µg/mL) led to nearly 100% of the cells being damaged. As is known, an S-shaped curve representing the dependence of cell damage on concentration or dose is associated with the heterogeneity of the cell population concerning resistance to the given exposure [96,97]. Macrophages, in turn, exhibit significant heterogeneity in various properties [98,99].

The mechanisms by which AgNP damage the cell membrane of eukaryotic cells remain insufficiently clear. However, based on our understanding and some of the literature data, the enhanced production of ROS caused by the presence of AgNP in the cell incubation medium, and the subsequent intensification of LPO in their membranes, may play a role in the membranotropic mechanism of the particles [60,67,100,101,102]. The more pronounced damage observed at an incubation temperature of 37 °C may indicate more intense LPO processes compared to those at 22 °C. Based on the results obtained and the literature data, we assume that the intensity of the LPO process in the lipid bilayer is influenced by the structural state of the bilayer [52,103,104].

Depending on the temperature, the lipid bilayer can exist in different structural states, such as gel or liquid crystal [53]. In the gel state, lipid molecules in the bilayer are packed more tightly, with fatty acid residues primarily occurring in the trans conformation and aligned parallel to each other, increasing the bilayer’s thickness and reducing its area. When the temperature rises to a certain “critical” value, the lipid bilayer transitions to a liquid crystal state. These temperature phase transitions are well-studied in artificial lipid membranes and model systems [105,106]. It has been found that the temperatures of such phase transitions depend on the lipid bilayer composition and can range from −70 to 80 °C depending on the lipid composition and the degree of fatty acid saturation [107,108,109]. In the liquid crystal phase, fatty acids undergo trans-gauche conformational transitions, causing the fatty acid chains to bend and disrupting their parallel alignment [103,104]. This leads to a decrease in bilayer thickness and an increase in its area. In this state, the membrane is more porous and more susceptible to ROS and LPO processes.

An earlier study investigated the effect of incubation medium temperature on the content of LPO products in erythrocyte membranes [57]. This study noted that in the lipid bilayer within the physiological temperature range of 0 to 40 °C, two structural conformational rearrangements occur, the first between 5–9 °C and 17–20 °C, leading to an intensification of LPO, and the second starting from 20–25 °C, further enhancing LPO. The more pronounced damage observed in our study at an incubation temperature of 37 °C with AgNP may be due to the intensification of LPO under these conditions.

Next, we studied the effect of incubation medium temperature in the range of 4–37 °C on cell membrane damage in the presence of AgNP at a final concentration of 2.5 µg/mL.

As seen in Figure 5, the dependence of the damaging effect of AgNP at a concentration of 2.5 µg/mL has a phase-like character. Increasing the ambient temperature from 4 °C to 22 °C leads to a decrease in the number of cells with damaged plasma membranes in the macrophage population. Further increasing the incubation medium temperature to 37 °C leads to an increase in the number of cells with damaged membranes. As mentioned earlier, at an incubation temperature of 4 °C, the lipid bilayer of the cell membrane is most likely in a gel state. This state of the bilayer is more compact, and the bilayer is less susceptible to LPO. However, under these conditions, almost 40% of the cell membranes are damaged. This significant damage appears to be related to the reduced activity of the cell’s AOD system enzymes at this temperature. This is particularly associated with the decreased activity, under these temperature conditions, of such important antioxidant enzymes as superoxide dismutase (SOD), catalase, and peroxidase [110].

With a subsequent increase in the macrophage incubation medium temperature with AgNP concentration kept constant the number of cells with damaged membranes decreases, which we believe is associated with the activation of the cell’s AOD enzyme system [111]. However, within the temperature range from 4 °C to 22 °C, as previously noted, a structural reorganization of the lipid bilayer into a liquid crystalline state occurs, making it more susceptible to LPO process.

The second observed phase on the curve is the phase of increased numbers of damaged cells when the cell incubation temperature is raised from 22 °C to 37 °C in the presence of AgNP. This is likely associated with the further intensification of the LPO process, which, as mentioned earlier, is driven by the subsequent phase reorganization of the lipid matrix and its transition to a liquid crystalline state, where the hydrophobic regions of lipids become more susceptible to LPO [52,103,104].

Next, we investigated the dependence of the damaging effects of AgNP on incubation time at incubation medium temperatures of 22 °C and 37 °C, with the anticipated more intensive LPO process initiated by ROS, which are presumably released by AgNP.

As seen in Figure 6, at an incubation temperature of 22 °C with silver nanoparticles at a concentration of 5 µg/mL, significant and reliable cell membrane damage is detected at 30 min, reaching approximately 20%. The figure shows that as the incubation time increases, the number of cells with damaged membranes also increases, reaching about 50% at 150 min of incubation. Further increases in incubation time are accompanied by a more pronounced increase in the number of cells with damaged membranes, up to 210 min, which may indicate the onset of “oxidative stress” after 150 min of incubation. Oxidative stress manifests as a significant accumulation of peroxide products in the lipid bilayer of cell membranes and the toxic effect of peroxide products on the cell’s AOD system. In this state, the cell’s AOD system fails to cope with the enhanced LPO process, leading to increased damage to the lipid bilayer of the membranes [56,97,112].

Increasing the incubation temperature to 37 °C with nanoparticles at the same concentration leads to significant damage of about 50% at 10 min of incubation, which reaches nearly 90% at 60 min of incubation, indicating an earlier onset of “oxidative stress” under these incubation conditions.

Thus, the study demonstrated that the damaging effect of silver nanoparticles on the plasma membranes of macrophages depends on their concentration, temperature, and exposure time. Increasing the concentration and incubation time leads to increased membrane damage, while changes in incubation temperature within the range of 4 to 37 °C can either enhance or reduce this effect depending on the state of the membrane’s lipid bilayer and the activity of the cell’s AOD system under different temperature conditions. These results led us to the necessity of constructing a mathematical model that accounts for the interaction between AgNP and cells, which we describe in the next section.

### 3.3. Mathematical Modeling of the Kinetics of a Diffusion-Controlled Reaction Between Mouse Macrophage Cells and AgNP in Solution

Let us consider the interaction between mouse macrophages and nanoparticles as consisting of two parallel reactions.

The first reaction involves the diffusion of the nanoparticle B towards the cell A with a rate constant $k_{1}$:(1)A+B→AB.

The second reaction involves the penetration of AgNP through the cell membrane or its adhesion to the surface, leading to the exposure of the cell to ROS and the activation of the AOD system—two opposing effects. This process can be schematically represented as follows:(2)$$\left(AB+AOD\right)+ROS\to dead\;cell.$$

Since the reactions proceed in parallel, we denote the total rate of both reactions leading to the formation of dead cells using the rate constant $k_{2}$. The use of the constant $k_{2}$ to describe the overall rate of both reactions is justified because the second reaction, being more complex and slower, is the determining step in the process of cell death and directly relates to the outcome—dead cells.

#### 3.3.1. Rate Constant of the First Reaction

To begin with, let us analyze the first reaction, which involves the interaction between a single silver nanoparticle and a macrophage. For this purpose, we will place the origin of the coordinate system at the center of the cell (particle A) (see Figure 7) and consider an ensemble of pairs of reacting particles, taking the concentration of particles B as the probability of finding a particle B at a distance r from the particle A.

The reaction rate is determined using the diffusion flux of this probability at the boundary of the reaction sphere with radius $R=r_{a}+r_{b}$.

We will assume that the influence of cells A on each other can be ignored, and around each cell A, it is possible to allocate a sphere with a sufficiently large radius where the probability of finding particles B is determined solely by the reaction (adhesion) with the cell A located at the center of this sphere.

We consider the following diffusion equation for this model in spherical coordinates:(3)$$\frac{\partial W}{\partial t}=D\left(\frac{\partial^{2}W}{\partial r^{2}}+\frac{2}{r}\frac{\partial W}{\partial r}\right).$$

By introducing ω=rW, the transformed equation is as follows:(4)$$\frac{\partial \omega }{\partial t}=D\frac{\partial^{2}\omega }{\partial r^{2}},$$
where $D=D_{A}+D_{B}$ is the “total” diffusion coefficient of particles A and B.

Let us solve Equation (4) using the Laplace transform method. The Laplace transform of the function $\omega \left(r,t\right)$ in time t is defined as follows:(5)$$L\left\{\omega \left(r,t\right)\right\}=\varpi \left(r,s\right)=\int_{0}^{\infty }\;e^{-st}\omega \left(r,t\right)dt,$$
where s is the complex variable of the Laplace transform, and $\varpi \left(r,s\right)$ is the image of the function $\omega \left(r,t\right)$ after the transformation.

Laplace transformations can be applied to both sides of the original Equation (4):(6)$$L\left\{\frac{\partial \omega }{\partial t}\right\}=L\left\{D\left(\frac{\partial^{2}\omega }{\partial r^{2}}\right)\right\},$$

The property of the Laplace transform is used as the time derivative:(7)$$L\left\{\frac{\partial \omega }{\partial t}\right\}=s\varpi \left(r,s\right)-\omega \left(r,0\right),$$
and, taking into account the initial condition $\omega \left(r,0\right)=rW_{0}$, we obtain the following:(8)$$s\varpi -rW_{0}=D\frac{\partial^{2}\varpi }{\partial r^{2}}.$$

The solution of Equation (8) has the following form:(9)$$\varpi =\frac{rW_{0}}{s}+C_{1}exp\left(-r\sqrt{\frac{s}{D}}\right)+C_{2}exp\left(r\sqrt{\frac{s}{D}}\right).$$

Let us find constants $C_{1}$ and $C_{2}$. To determine these constants, we use the initial and boundary conditions and solve the problem in the “black sphere” approximation for a uniform distribution of particles B:(10)$$W\left(r,0\right)=W_{0},\;\;\omega \left(r,0\right)=rW_{0},$$(11)$$W\left(R,t\right)=0,\;\;\;\omega \left(R,t\right)=rW_{0},$$(12)$$\;W\left(\infty ,t\right)=W_{0},\;\;\omega \left(r,0\right)=rW_{0}.$$

For this purpose, we apply the Laplace transform to the boundary conditions (10) for function $\varpi \left(r,s\right)$:(13)$$\varpi \left(R,s\right)=0,$$(14)$$\;\varpi \left(\infty ,s\right)=\frac{rW_{0}}{s}.$$

Condition (14) requires boundedness of the solution for the function ϖ from Equation (9). Hence, we obtain $C_{2}=0$. Given condition (13), solution (9) will take the following form:(15)$$\varpi =\frac{rW_{0}}{s}-\frac{RW_{0}}{s}exp\left(-\left(r-R\right)\sqrt{\frac{s}{D}}\right).$$

After the inverse Laplace transform for the expression (15), we obtain(16)$$W\left(r,t\right)=W_{0}\left(1-\frac{R}{r}Erfc\left(\frac{r-R}{2\sqrt{Dt}}\right)\right),$$
where $Erfc\left(x\right)=1-Erf\left(x\right)$ is the complementary error function. Solution (16) describes the time evolution of the distribution of particles B around cell A (see Figure 8).

Initially, the uniform distribution will change due to the reaction of the closely located pairs. Subsequently, the reaction rate will slow down as the pairs of reactants with larger distances start reacting, as the diffusive approach requires time. After some time, a stationary probability distribution W will be established.

Using the probability W(r,t) from Equation (16), we can find the rate constant for the reaction. The reaction rate per cell A is equal to the total diffusive flux j across boundary R:(17)$$j=4\pi R^{2}D\left(\frac{\partial W}{\partial r}\right){|}_{r=R}=4\pi RDW_{0}\left(1+\frac{R}{\sqrt{\pi Dt}}\right),$$
where $W_{0}$ is the concentration of particles B. Then,(18)$$W_{r}=4\pi RD\left(1+\frac{R}{\sqrt{\pi Dt}}\right)\left[A\right]\left[B\right],$$
where the rate constant is derived as follows:(19)$$k_{1}=4\pi RD\left(1+\frac{R}{\sqrt{\pi Dt}}\right).$$

In the experiment, the initial section of the graph, depicting cell death over time (Figure 7) with an infinitely high cell death rate (as t→0), is not observed. Under normal conditions, the contribution of the non-stationary term (dependent on time) to the kinetic reaction curve is negligible, especially considering the relatively long duration of the experiment. Therefore, we will focus only on the stationary part:(20)$$k_{1}=4\pi RD.$$

Let us determine the diffusion coefficient D through the Stokes–Einstein model for spherical particles [85,87]:(21)$$D=\frac{k_{B}T}{6\pi \eta }\left(\frac{1}{r_{A}}+\frac{1}{r_{B}}\right),$$
where $k_{B}$ is the Boltzmann constant, η is the viscosity of the saline solution, T is the temperature in kelvin, and the dimensionality of the constant is $m^{3}$/s.

#### 3.3.2. Rate Constant of the Second Reaction

In the absence of antioxidants and antioxidant enzymes, or in the case of their dysfunction, the cell is guaranteed to die upon interaction with nanoparticles due to the generation of ROS by the particles. Let the effective energy (activation energy) required for cell death be $E_{act}^{1}$ for the first reaction (without considering the AOD system). Then, $E_{act}^{2}$ represents the energy required for cell death in the presence of AOD activity.

Suppose a cell had no antioxidant defense; encountering silver nanoparticles would be fatal. We can then assume that the presence of a single silver nanoparticle in the first reaction initiates the reaction. Let us write the expressions of the activation energies for the first and second reactions, respectively:(22)$$E_{act}^{1}=-k_{b}Tln\left(\frac{k_{1}}{A_{1}}\right),$$(23)$$E_{act}^{2}=-k_{b}Tln\left(\frac{k_{2}}{A_{1}}\right),$$
where $k_{1}$ and $k_{2}$ are the reaction rate constants we found earlier.

In this case, suppose that $\frac{E_{act}^{2}}{E_{act}^{1}}=\frac{P_{1}}{P_{2}};P_{1}=100\%,P_{2}=P\left(T\right),$ where $P_{1}\left(T\right)$ and $P_{2}\left(T\right)$ are the probabilities of cell death at temperature *T* under the influence of surrounding silver particles in the first and second reactions, respectively. The probability $P_{2}$ represents the contribution of AOD of the cell at different temperatures.

It follows that(24)$$E_{act}^{2}=\frac{E_{act}^{1}}{P_{2}\left(T\right)},$$

From Equations (22) and (23), we obtain(25)$$\frac{k_{1}}{k_{2}}=\frac{Aexp\left(\frac{-E_{act}^{1}}{k_{b}T}\right)}{Aexp\left(\frac{-E_{act}^{2}}{k_{b}T}\right)}.$$

Then, from Equations (24) and (25), we have(26)$$k_{2}=\frac{k_{1}}{exp\left(E_{act}^{1}\left(\frac{1-P\left(T\right)}{k_{b}TP\left(T\right)}\right)\right)}.$$

Let the activation energy $E_{aKT}^{1}\;$be equal to the average kinetic energy of the silver particle:(27)$$\bar{V}=\sqrt{\frac{3k_{b}T}{m}};\;E_{act}^{1}=\frac{m{\bar{V}}^{2}}{2}=\frac{3k_{b}Tm}{2m}=\frac{3}{2}k_{b}T.$$

As a result, from Equations (20), (21), (26) and (27), we obtain an expression for $k_{2}$:(28)$$k_{2}=\frac{4\pi \left(r_{A}+r_{B}\right)\frac{k_{b}T}{6\pi \eta }\left(\frac{1}{r_{A}}+\frac{1}{r_{B}}\right)}{exp\left(\frac{3-{3P}_{2}\left(T\right)}{{2P}_{2}\left(T\right)}\right)},$$
where $P_{2}\left(T\right)$ is the probability of cell death under the influence of AgNP as a function of temperature; the probabilities will be estimated using our experimental data (Figure 6 and Figure 7).

#### 3.3.3. Rate of AB Complex Formation

Knowing the rate constant of the second reaction, we estimate the cell death rate. Consider, in the following equation, that a large number of nanoparticles can be deposited on the cell simultaneously. Since the cell is much larger than the nanoparticles, the change in its effective radius due to the adsorption of nanoparticles will be relatively small. Thus, the change in the diffusion rate of the complex due to the increase in its size will be considered insignificant against the background of the total size of the cell.

Let us use $N\left(t\right)$ to denote the number of dead cells as a function of time. Then, we have the following equation:(29)$$k_{2}\left(C_{A_{0}}-N\left(t\right)\right)\left(C_{B_{0}}-\frac{C_{B_{0}}}{C_{A_{0}}}N\left(t\right)\right)=\frac{dN\left(t\right)}{dt},$$
where $C_{A_{0}}$ and $C_{B_{0}}$ are the initial concentrations of cells and silver nanoparticles, respectively, with the dimensions of their concentrations shown in particles/m^3^.

Using the initial condition $N\left(0\right)=0$, we can find the following expression for $N\left(t\right)$:(30)$$N\left(t\right)=\frac{C_{A_{0}}C_{B_{0}}k_{2}t}{C_{B_{0}}k_{2}t+1}.$$

#### 3.3.4. Numerical Results

We used the following parameters for the calculations:Nanoparticle concentration ($C_{NP})\;$5 or 2.5 μg/mL;Bulk density (ρ)
of silver nanoparticles 350 kg/m^3^;Macrophage radius ($r_{MP}$) $5\times 10^{-6}$ m;Average radius of the silver particle ($r_{NP}$) $35\times 10^{-9}$ m;Macrophage count ($C_{MP}$) $5\times 10^{5}$ units/mL.

Since Equation (30) assumes that variables in the SI system and concentrations should be expressed as particles per cubic meter (particles/m^3^), we convert the nanoparticle concentrations from μg/mL to this unit using the following equation:$$C_{NP}[\frac{\mathrm{particles}\;}{m^{3}}]=\frac{C_{NP}[\frac{\mu g}{mL}]}{1000\rho \frac{4}{3}\pi r_{NP}^{3}}$$

The viscosity of the solution was taken to be equal to the viscosity of water as a function of temperature, calculated using the following formula (Figure 9):(31)$$\eta =A\times 10^{\frac{B}{\left(T-C\right)}},$$
where $A=2.414\times 10^{-5}\;Pa\cdot s;\;B=247.8\;K;\;C=140\;K$; T is the temperature in kelvin.

Next, we compare our experimental data on the dependence of the relative number of damaged cells on incubation time with our theoretical estimates.

Figure 10 shows that our kinetic-diffusion model at 22 °C can predict cell death well at exposure times of up to 150 min and, at 37 °C, the model can predict cell death at times of up to 210 min. When using an incubation medium temperature of 22 °C, a divergence of experimental and the theoretical data is observed when the incubation time is increased beyond 150 min, which is hypothesized to be associated with the onset of oxidative stress [56,97,112].

To assess the accuracy and adequacy of the proposed model, we used the coefficient of determination $R^{2}$, which is a statistical measure reflecting the proportion of variation in the dependent variable explained by the model compared to the total variation.

The coefficient of determination is calculated by the following formula:(32)$$R^{2}=1-\frac{SS_{res}}{SS_{tot}},$$
where $SS_{res}$ is the sum of the squares of differences between the experimental and model values (residual sum of squares), and $SS_{tot}$ is the total sum of squares of differences between the experimental values and their mean (total sum of squares). The value $SS_{res}$ is calculated as follows:(33)$$SS_{res}=\sum_{i=1}^{n}\;{\left(y_{i}-{\hat{y}}_{i}\right)}^{2},$$
where $y_{i}$—experimental values; ${\hat{y}}_{i}$—values predicted by the model. In turn, $SS_{tot}$ is determined by the following expression:(34)$$SS_{tot}=\sum_{i=1}^{n}\;{\left(y_{i}-\bar{y}\right)}^{2},$$
where $\bar{y}$ is the average value of the experimental data.

The results of the calculations showed the coefficients of determination were $R^{2}=0.893$ for temperature 22 °C and $R^{2}=0.997$ for temperature 37 °C, indicating a high degree of agreement between the model predictions and experimental data.

These $R^{2}$ values indicate that the proposed model adequately describes the processes occurring during the interaction of AgNP with cells and can be used to predict the effects of cell death under various conditions.

Next, we calculated the activation energy $E_{act}^{2}$ we discussed above (Equation (23)). From Figure 11, it is evident that the activation energy required for cell death is maximal at a temperature range of 20–25 °C. This indicates that, within this temperature interval, the cells are least susceptible to the detrimental effects of nanoparticles.

Our theoretical approach shows high convergence with the experimental data, where the minimum damage was observed at 22 °C (Figure 4). There is a particular correlation with the results of experiments at relatively low temperatures (4 °C). The activation energy of the reaction at the given temperature, which determines the interaction between the cell and AgNP, is minimal, making collisions between the cell and AgNP particularly harmful to the cell, as observed in the experiment. It should also be noted that under such temperature conditions, the activity of the enzymatic AOD systems is negligible compared to their activity at higher temperatures. Additionally, the lower activation energy of the reaction at 37 °C compared to that at 22 °C theoretically predicts an increase in the number of cells with damaged membranes at 37 °C, which was also observed in the experiment.

Thus, the proposed approach based on a combination of diffusion-controlled reaction theory and cell death kinetics due to oxidative stress can be used to predict and minimize the toxic effects of nanoparticles under different conditions.

The developed model, despite its relative simplicity, takes into account the main processes of the interaction of nanoparticles with cells and satisfactorily describes the accumulated experimental data on the kinetics of cell death. In the future, we plan to improve the model by providing a more detailed description of the mechanisms of ROS generation, AOD operation, and cell death.

### 3.4. Influence of Pro- and Antioxidant Factors on the Toxic Effect of AgNP

To deepen the understanding of the membranotropic action of AgNP on macrophage plasma membranes, we studied the change in the intensity of this effect under the influence of pro- and antioxidant factors in the presence of nanoparticles. Serotonin was used as an antioxidant at a final concentration of 1 mg/mL. As a prooxidant, medium-wave ultraviolet radiation (UV-B) with a wavelength of λ_max_ = 306 nm and a radiation dose of 5 J/cm^2^ was applied.

As seen in Figure 12, incubating macrophages with AgNP at a final concentration of 5 µg/mL at 22 °C for 60 min results in approximately 30% of the macrophage population having damaged membranes. In turn, UV-B radiation at a dose of 5 J/cm^2^ results in 25% of cells having damaged membranes. The membranotropic action of UV radiation within this specific spectral range was repeatedly utilized in our prior experimental studies [113]. UV exposure was observed to intensify the LPO processes and induce structural defects in the lipid bilayer of macrophage plasma membranes, resulting in a marked increase in cells with compromised membrane integrity. However, combined exposure to nanoparticles (5 µg/mL) and UV radiation at the same dosage elicited a non-additive interaction, producing membrane damage exceeding the arithmetic sum of individual treatments. This synergistic effect led to approximately 80% of cells exhibiting membrane impairment. The observed synergy arises from UV-B radiation acting as a potent inducer of LPO within membrane lipid bilayers, thereby amplifying bilayer defects. Nanoparticles likely exacerbate this process through catalytic surface interactions or photodynamic enhancement mechanisms [114,115]. We hypothesize that AgNP further promotes the accumulation of LPO products through ROS generation. The combined action of these two stressors potentiates damage to the macrophage plasma membrane lipid bilayers via this dual-oxidative pathway. The observed synergistic effect (UV-B + AgNP) strongly suggests that the membranotropic activity of AgNP is mechanistically linked to their capacity to induce ROS.

Simultaneous exposure to serotonin at a final concentration of 1 mg/mL and AgNP at a final concentration of 5 µg/mL significantly and reliably reduces the membranotropic effect of AgNP compared to that observed in the absence of serotonin. Serotonin is believed to be a radioprotector and reduces the effects of ROS and the LPO process [116,117,118], which leads to a reduction in the damaging effect.

These experiments may indicate a significant role of ROS and the lipid peroxidation process in the observed damaging effects of AgNP on the plasma membranes of macrophages. To further confirm the involvement of ROS and the LPO process in the toxic effect, the following experiments were conducted.

### 3.5. Study of the Effect of AgNP on ROS Levels in Cell Preparations and Cell Suspensions

Figure 13 presents data illustrating the effect of AgNP at final concentrations of 1, 2.5, and 5 μg/mL on ROS generation over 210 min at 22 °C. This effect was assessed based on the fluorescence of the accumulated fluorescent dye DCFH-DA in cells and compared to the control group without nanoparticles.

Figure 13 demonstrates that AgNP induces a dose-dependent increase in ROS generation by macrophages compared to the control, which continues to rise over the course of 210 min. As positive controls, the kinetics of samples treated with the ROS inducer PMA at a minimal concentration of 20 nM and hydrogen peroxide at a final concentration of 30 mM were used. The minimal concentration of PMA caused an increase in ROS generation by macrophages compared to the control, serving as a positive test for the functional response of macrophages to this stimulus.

Hydrogen peroxide, added to the incubation medium at the specified concentration, significantly increased ROS generation compared to the control. In this case, the ROS levels were higher than those observed with AgNP at the maximum tested concentration.

A similar pattern was observed when using the same agents at the same concentrations, but at an incubation temperature of 37 °C (Figure 14).

The figure demonstrates that AgNP also induces a dose-dependent increase in ROS generation by macrophages compared to the control, with levels continuing to rise over 210 min. PMA and hydrogen peroxide similarly enhance ROS generation by macrophages; however, at 37 °C, the values for all corresponding experimental points in the kinetics are higher than those observed at 22 °C, including the control values.

These results provide strong evidence supporting the previously stated hypothesis regarding the key role of ROS in initiating damage to macrophage cellular membranes via AgNP.

### 3.6. Investigation of AgNP Effects on Macrophage Membranes via Scanning Electron Microscopy

We further examined the impact of AgNP on macrophage membrane morphology following a 60 min incubation period at 22 °C and subsequent fixation procedures (Figure 15).

As shown in the Figure 15, a 60 min exposure of macrophages to AgNP at a final concentration of 5 µg/mL (22 °C) induces a pronounced disruption of the plasma membrane (Figure 15b). As demonstrated in prior experimental studies, this membrane destabilization correlates with intensified LPO of the bilayer structure, a consequence of ROS generation triggered by nanoparticle–cell interactions.

## 4. Conclusions 

In this study, we carried out an investigation into the damaging effects of AgNP on the plasma membranes of mouse peritoneal macrophages. The results demonstrated that the extent of membrane damage depends on the concentration of nanoparticles, incubation time, and temperature. Higher concentrations and prolonged exposure to AgNP lead to increased damage to cell membranes. The dependence of AgNP’s damaging effect on macrophage membranes on the incubation temperature exhibits a phase-like character, with more pronounced membrane damage observed at 4 °C and 37 °C compared to 22 °C. The influence of the pro-oxidant factor (UV-B) enhances the degree of plasma membrane damage, while the antioxidant (serotonin) reduces this effect.

The obtained experimental data indicate a significant role of ROS and the associated LPO process in the damaging effects of nanoparticles on the plasma membranes of macrophages. Additionally, the state of the lipid bilayer of cellular membranes and the activity of AOD systems are crucial factors.

In this work, a mathematical model was also developed and applied to describe the kinetics of the interaction between AgNP and macrophages. This model, based on the theory of diffusion-controlled reactions and the kinetics of cell death due to oxidative stress, showed a high degree of correspondence with the experimental data. The results of the equations of the mathematical model also support the hypothesis regarding the key role of ROS and LPO processes in the damage of cellular membranes via silver nanoparticles.

The cellular model system based on mouse peritoneal macrophages used in this study proved to be convenient for studying and understanding the effects of nanoparticles, allowing for a detailed analysis at the cellular level. This approach can be useful for developing methods to assess risks and create safe nanomaterials.

Future research in this area should focus on several key points to advance the understanding and safe application of AgNP. Firstly, it is essential to combine experimental studies with theoretical models to create a comprehensive approach to studying nanoparticle toxicity. Improving the mathematical model by including a more detailed description of the mechanisms of ROS generation and the function of AOD in cells will enhance the model’s predictive capability and its applicability under various conditions and with different types of nanoparticles. Conducting direct experiments to determine the influence of cellular AOD system activity on the cell’s resistance to AgNP exposure and comprehensive in vivo studies to assess the systemic toxicity and distribution of AgNP in the body will provide a better understanding of the potential health risks of prolonged nanoparticle exposure. Additionally, investigating the long-term effects of AgNP on living systems at various levels of organization will help evaluate their cumulative effects and potential for bioaccumulation. Optimizing and studying the characteristics of AgNP, such as size, shape, and surface modifications, can help reduce their toxicity while maintaining their beneficial properties. Using various methods and approaches to study the mechanisms of nanoparticle action in greater detail will also be crucial. Furthermore, investigating the environmental impacts of AgNP usage, including effects on aquatic and terrestrial ecosystems, is important for understanding the widespread risks associated with nanomaterial application. Developing standards and principles for the safe use of AgNP in consumer products and medical applications will ensure public health and safety. These directions for future research will help create safer and more effective nanomaterials, balancing their beneficial properties with their potential health and environmental risks.

## Figures and Tables

**Figure 1 pharmaceutics-17-00398-f001:**
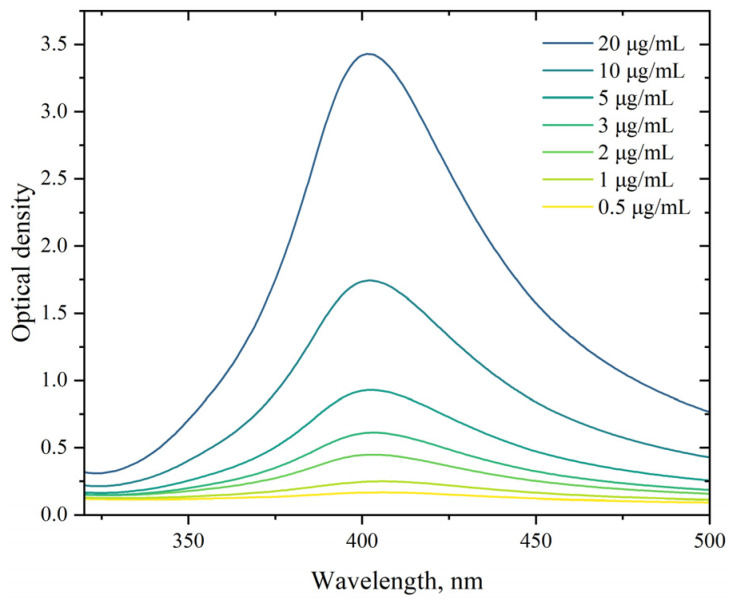
Absorption spectra of AgNP. The horizontal axis represents the wavelength, nm; the vertical axis shows the optical density.

**Figure 2 pharmaceutics-17-00398-f002:**
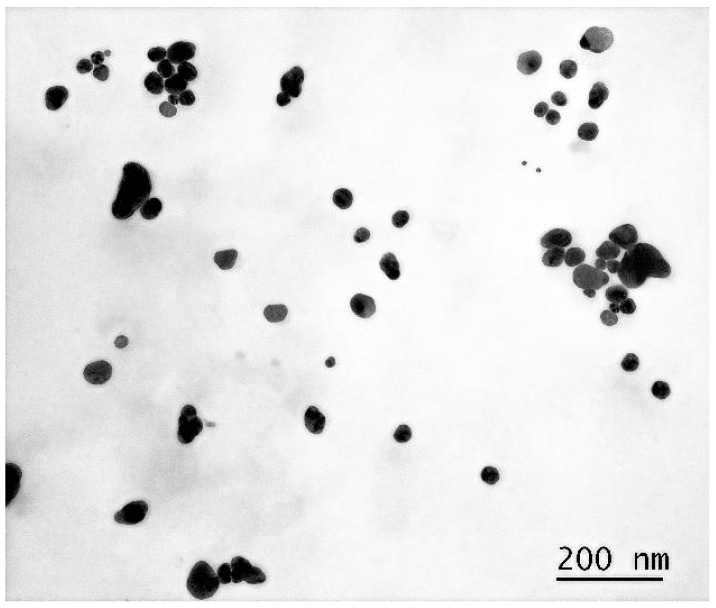
Micrograph of a dried droplet of AgNP suspension on a substrate at 230,000× magnification.

**Figure 3 pharmaceutics-17-00398-f003:**
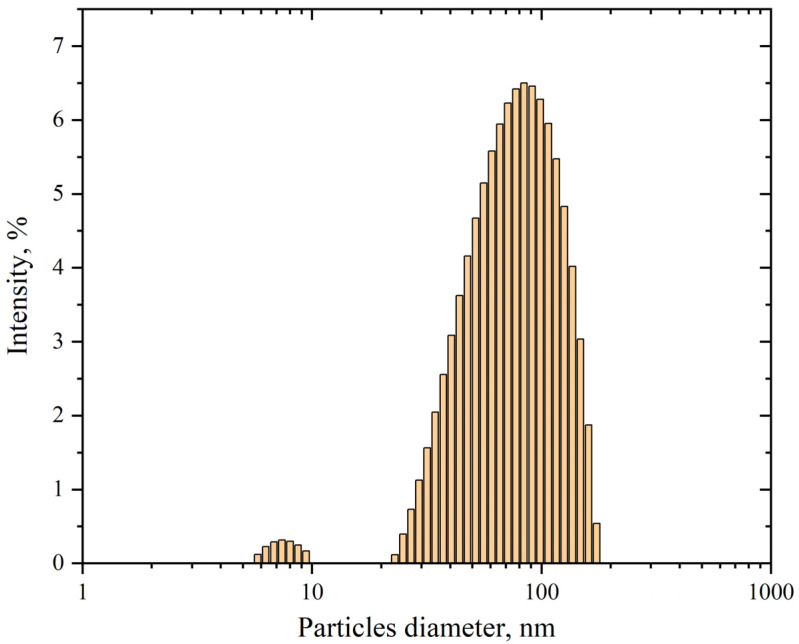
Size distribution of AgNP in suspension. The horizontal axis represents the particles diameter, nm; the vertical axis shows the intensity, %. The polydispersity is 24.4%.

**Figure 4 pharmaceutics-17-00398-f004:**
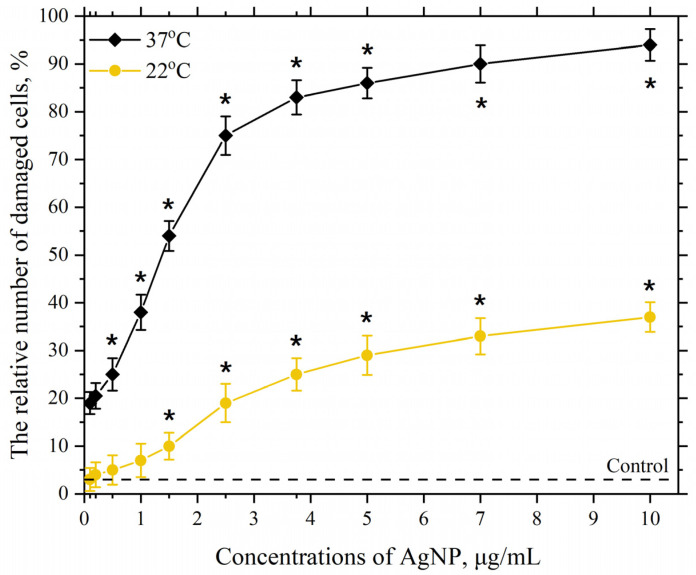
Damaging effect of AgNP on the plasma membranes of mouse peritoneal macrophages. The horizontal axis represents the concentration of AgNP in the macrophage incubation medium, µg/mL; the vertical axis shows the relative content of cells with damaged plasma membranes in the macrophage population, %. The incubation of cells with AgNP was carried out for 60 min at 22 °C and 37 °C; *—statistically significant values compared to the control (*p* < 0.05).

**Figure 5 pharmaceutics-17-00398-f005:**
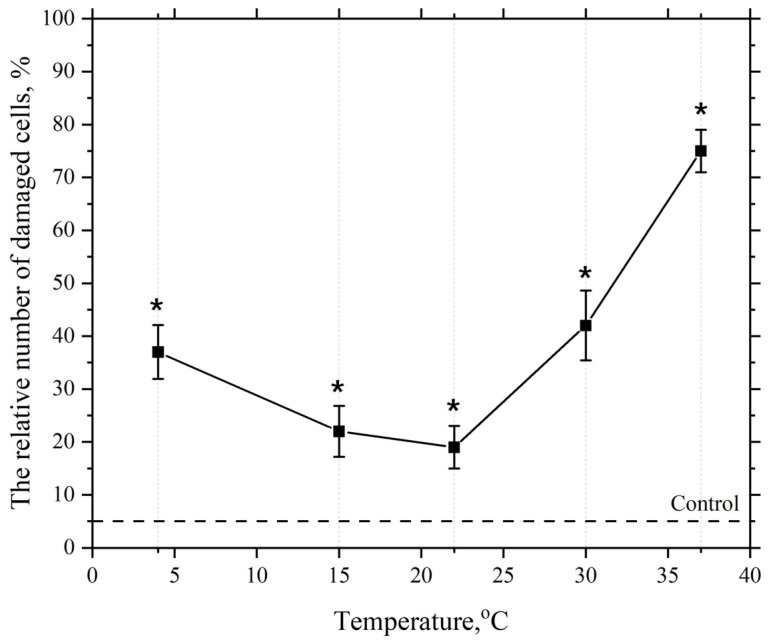
Dependence of the damaging effect of AgNP on the plasma membranes of mouse peritoneal macrophages on temperature. The horizontal axis represents the temperature of the macrophage incubation medium, °C; the vertical axis shows relative content of cells with damaged plasma membranes in the macrophage population, %. The cells were incubated with AgNP at a final concentration of 2.5 µg/mL for 60 min; *—statistically significant values compared to the control (*p* < 0.05).

**Figure 6 pharmaceutics-17-00398-f006:**
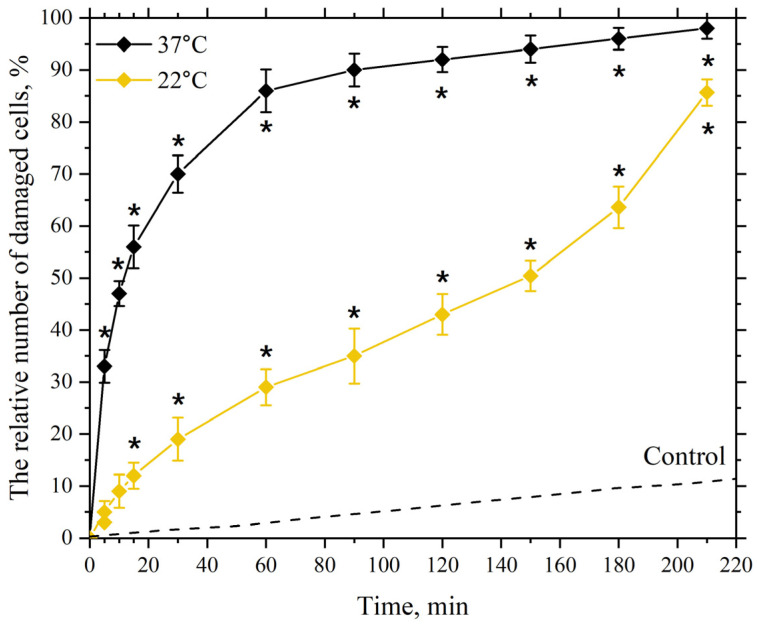
Dependence of the damaging effect of AgNP on the plasma membranes of mouse peritoneal macrophages on incubation time. The horizontal axis represents the incubation time of macrophages with AgNP, min; the vertical axis shows the relative content of cells with damaged plasma membranes in the macrophage population, %. The cells were incubated with AgNP at a final concentration of 5 µg/mL at temperatures of 22 °C and 37 °C; *—statistically significant values compared to the control (*p* < 0.05).

**Figure 7 pharmaceutics-17-00398-f007:**
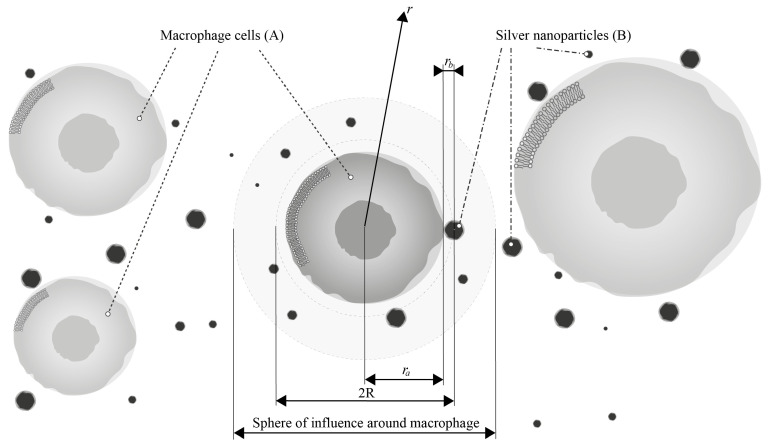
Kinetic model of the interaction of AgNP (**B**) with macrophages (**A**), where $r_{a}$ is the radius of macrophages and $r_{b}$ is the radius of nanoparticles.

**Figure 8 pharmaceutics-17-00398-f008:**
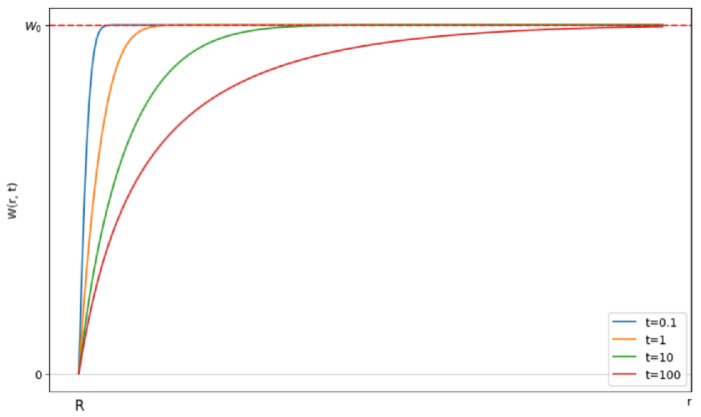
Evolution of the probability W(r,t) of finding a particle B in the vicinity of the cell A over time. The horizontal axis represents the distance *r* from the cell surface. The vertical axis shows the probability $W\left(r,t\right)$ of finding a silver nanoparticle at distance r at time t.

**Figure 9 pharmaceutics-17-00398-f009:**
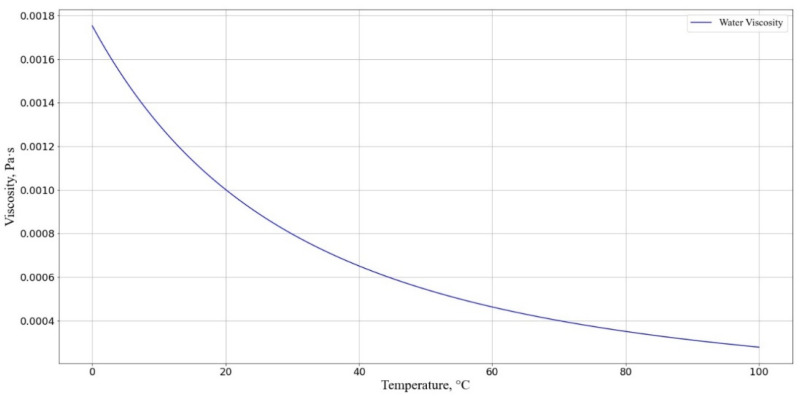
Dependence of solution viscosity η(T) at different temperatures. The horizontal axis represents the cell incubation medium temperature, °C; the vertical axis shows the solution viscosity η.

**Figure 10 pharmaceutics-17-00398-f010:**
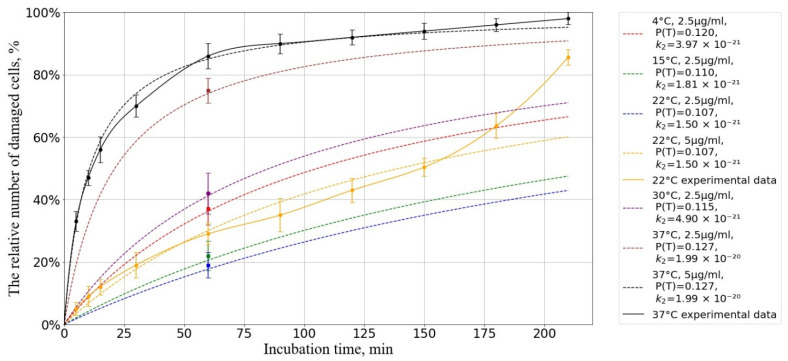
Dependence of the number of damaged cells on the incubation time at different temperatures. The horizontal axis represents the incubation time, min; the vertical axis shows the relative number of cells with a damaged membrane, %. The initial concentration of AgNP is $B_{0}$ = $3.98\;\times \;10^{16}$ particles/m^3^. Experimental results (solid lines) and forecast (dotted lines).

**Figure 11 pharmaceutics-17-00398-f011:**
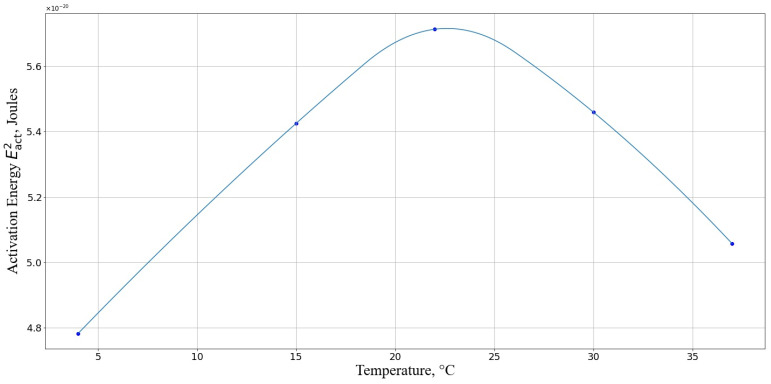
Activation energy $E_{act}^{2},$ required for macrophage death for different temperatures. The horizontal axis represents the temperature, °C; the vertical axis shows the activation energy in Joules.

**Figure 12 pharmaceutics-17-00398-f012:**
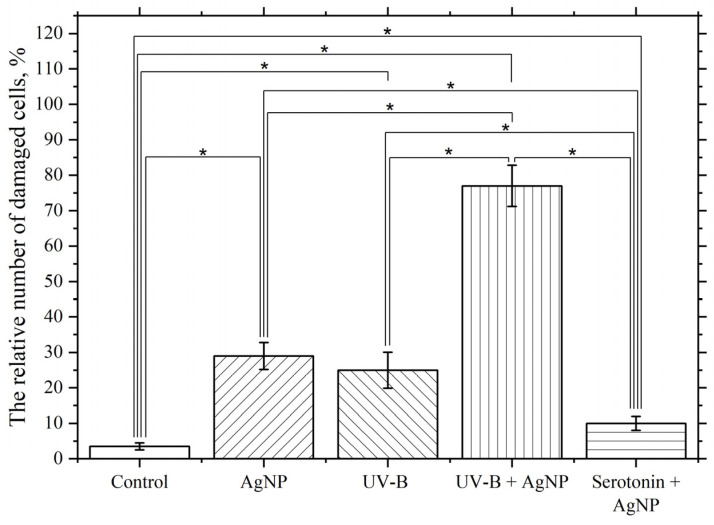
Influence of UV-B and serotonin on the damaging effect of AgNP. The vertical axis represents the relative content of cells with damaged plasma membranes in the macrophage population, %. The effect of UV radiation on macrophages was studied at a dose of 5 J/cm^2^ with a wavelength of λmax = 306 nm (UV-B). The cells were incubated at 22 °C with AgNP at a final concentration of 5 µg/mL, without additional factors (AgNP), with simultaneous UV radiation (UV-B + AgNP), and with simultaneous serotonin at a final concentration of 1 mg/mL (serotonin + AgNP); *—statistically significant differences compared to the control and between experimental groups (*p* < 0.05).

**Figure 13 pharmaceutics-17-00398-f013:**
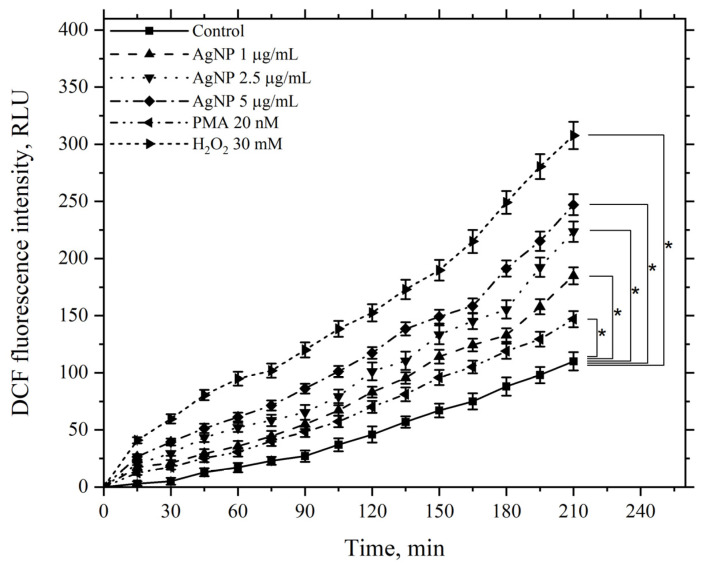
Kinetics of the ROS levels produced by macrophages during incubation with AgNP, hydrogen peroxide, and PMA. The horizontal axis represents time, min; the vertical axis indicates the fluorescence intensity of DCFH-DA, RLU. Macrophages were incubated with the agents at a temperature of 22 °C; *—statistically significant values related to the entire curve on the graph in comparison to the control curve (*p* < 0.05).

**Figure 14 pharmaceutics-17-00398-f014:**
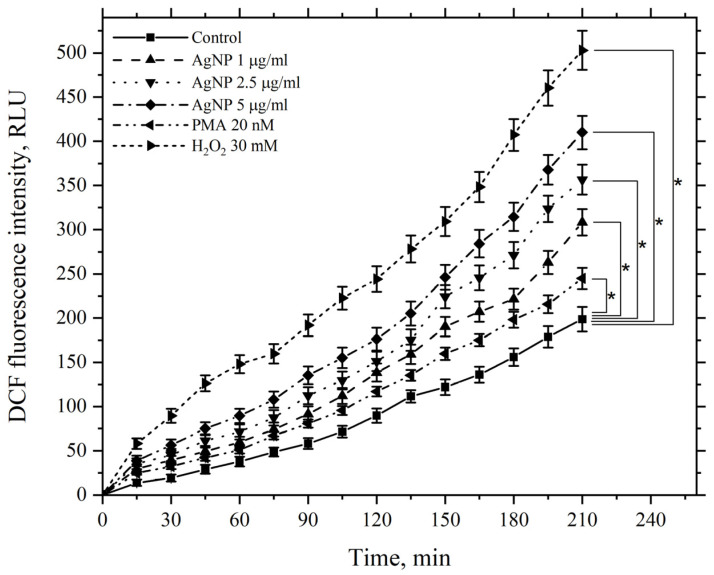
Kinetics of ROS levels produced by macrophages during incubation with AgNP, hydrogen peroxide, and PMA. The horizontal axis represents time, min; the vertical axis shows the fluorescence intensity of DCFH-DA, RLU. Macrophages were incubated with the agents at a temperature of 37 °C; *—statistically significant values related to the entire curve on the graph in comparison to the control curve (*p* < 0.05).

**Figure 15 pharmaceutics-17-00398-f015:**
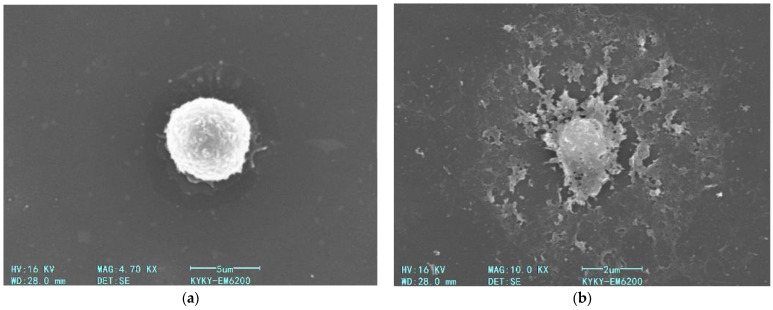
Scanning electron micrographs of fixed macrophages: (**a**) control cells incubated without AgNP; (**b**) cells after 60 min incubation with AgNP (5 µg/mL) at 22 °C.

## Data Availability

Data are contained within the article.

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
