# Peer review of "Investigation of Cell Damage Induced by Silver Nanoparticles in a Model Cell System"

_pharmaceutics, 2025, doi:10.3390/pharmaceutics17040398_

Round 1
Reviewer 1 Report
Comments and Suggestions for Authors
Dear authors,
I congratulate you for your good investigation. You got an excellent manuscript, with important conclusions and it is written with a strong pedagogic vision.
All the best.
Reviewer 2 Report
Comments and Suggestions for Authors
The authors present a mathematical model describing the relationship between silver nanoparticle concentration and the destruction of cell membranes.
Overall, the article is well presented, although there are aspects that require further elaboration or improvement.
It would be necessary to explain the choice of the cellular model—why were peritoneal macrophages specifically selected? Considering that these cells are generally circulating and exhibit adhesion capacity in culture, do the authors consider this to be the correct cellular model for developing this type of mathematical model?
Why do the authors use dried-air incubation? Doesn't the absence of COâ‚‚ or a regulated control of gas exchanges impact cell viability?
Regarding the experiments, the control line is mentioned, but it is unclear what conditions are considered as controls. This should be clarified in the text.
In the experimental section, details are missing regarding the concentrations and media used for diluting the nanoparticles for SEM, TEM, and DLS measurements, as well as the method used for nanoparticle quantification.
In the results section, some lexical adjustments are needed. The term "coagulation" is not the most appropriate when referring to nanoparticles; it should be replaced with "aggregation" or "sedimentation," depending on the context.
The SEM image is not informative: nanoparticles of 200 nm are measured using a 4-micron scale, which does not allow for proper visualization of their surface. The authors should provide a micrograph at higher magnifications; if the SEM used does not allow for this, the SEM section should be omitted as it is not informative.
The authors mention observing aggregation in SEM images. However, this is not indicative of actual aggregation in the culture medium but rather a common artifact of the technique.
The authors correlate TEM measurements with DLS measurements, but these techniques do not analyze the same aspect: one focuses on the core of the nanoparticles, while the other examines their hydrodynamic radius in the solution. In vitro or in vivo, the latter is more significant in interactions with membranes. The authors should discuss this further, specifying the medium in which the measurements were taken.
Regarding the developed mathematical model, macrophage size is considered assuming a spherical shape—is this correct? While it is understandable as a necessary approximation for calculations, the authors should discuss whether this mathematical model could also be applied to other cell types.
Finally, the purpose of UV irradiation is unclear. Is it an additional factor to study its effects on the membrane? The authors should further elaborate on this aspect.
Reviewer 3 Report
Comments and Suggestions for Authors
In this investigation, cytotoxicity of the silver nanoparticles on macrophage membranes has been studied with different optimizing parameters. This is an important piece of work which will help to explore new methods in the medicinal application of Nano-silver. The MS should be revised with the following comments
1.In this manuscript silver nanoparticles abbreviated a "NpAg" It could be written as AgNp.
2. In the abstract, the size range of the synthesized nps should be included. In the results and discussion, please indicate the stability of the nanoparticles (number of days/period).
3. What is the significance in selecting wide range of incubation temperature (4 to 37 degree C), should be incorporated in the appropriate section
4. In this experiments, the cytotoxicity was analyzed by light scattering, UV-B irradiation, ROS determination methods. By using mathematical model, the interactions were studied. Have you observed the morphological changes/ membrane damages on the treated macrophages through microscopic examination? If you have any data please include it in the discussion.
5. Figure 2 and figure 3 should be replaced with clear and sharper image which should represent clarity of the nanoparticles.
6. fig. 5,6,7,11,13,14 &15 - indicate the significance of the error bar in the figure caption.
Round 2
Reviewer 2 Report
Comments and Suggestions for Authors
The reviewer thanks the authors for the changes and for clarifying all the points that needed to be addressed after the first round of revision.